# Endocytoscopic Observation of Esophageal Lesions: Our Own Experience and a Review of the Literature

**DOI:** 10.3390/diagnostics12092222

**Published:** 2022-09-14

**Authors:** Youichi Kumagai, Kaiyo Takubo, Kenro Kawada, Masayuki Ohue, Morihiro Higashi, Toru Ishiguro, Satoshi Hatano, Yoshitaka Toyomasu, Takatoshi Matsuyama, Erito Mochiki, Hideyuki Ishida

**Affiliations:** 1Department of Digestive Tract and General Surgery, Saitama Medical Center, Saitama Medical University, Kawagoe 350-8550, Saitama, Japan; 2Research Team for Geriatric Pathology, Tokyo Metropolitan Institute of Gerontology, Tokyo 173-0015, Japan; 3Department of Esophageal and General Surgery, Tokyo Medical and Dental University, Tokyo 113-8510, Japan; 4Department of Surgery, Osaka International Cancer Center, Osaka 541-8567, Japan; 5Department of Pathology, Saitama Medical Center, Saitama Medical University, Saitama 350-0495, Japan

**Keywords:** endocytoscopy, esophagus, artificial intelligence, esophageal cancer, esophagitis, vital staining

## Abstract

This review outlines the process of the development of the endocytoscope (EC) with reference to previously reported studies including our own. The EC is an ultra-high-magnification endoscope capable of imaging at the cellular level. The esophagus is the most suitable site for EC observation because it is amenable to vital staining. The diagnosis of esophageal lesions using EC is based on nuclear density and nuclear abnormality, allowing biopsy histology to be omitted. The observation of nuclear abnormality requires a magnification of ×600 or higher using digital technology. Several staining methods have been proposed, but single staining with toluidine blue or methylene blue is most suitable because the contrast at the border of a cancerous area can be easily identified. A three-tier classification of esophageal lesions visualized by EC is proposed: Type 1 (non-cancerous), Type 2 (endocytoscopic borderline), and Type 3 (cancerous). Since characteristic EC images reflecting pathology can be obtained from non-cancerous esophageal lesions, a modified form of classification with four additional characteristic non-cancerous EC features has also been proposed. Recently, deep-learning AI for analysis of esophageal EC images has revealed that its diagnostic accuracy is comparable to that of expert pathologists.

## 1. Introduction

The endocytoscope (EC) is a flexible ultra-high-magnification endoscope that can be inserted into the digestive tract [1]. It has a magnification of ×500–1000, which is five to ten times higher than that of a conventional magnifying endoscope (×100). Although a standard magnifying endoscope allows observation of surface microvasculature or mucosal pits [2,3] and is useful for early detection of gastrointestinal cancer and diagnosis of invasion depth, it is unable to visualize individual cells directly at ×100, which precludes a final histological diagnosis. The EC, on the other hand, allows observation of surface epithelial cells in vivo in real time, in tandem with vital staining.

The history of development of the EC began in 2000 when Ohue, one of our coauthors [4], performed ultra-high-magnification studies of resected colorectal cancer specimens using the contact endoscopy technique developed by Karl Storz. He succeeded in visualizing colorectal cancer cells ex vivo. We have studied esophageal endoscopy and pathology for a long time. Therefore, we considered that this technique was much more useful in the esophagus than in the colon, because nuclear density and morphology (size and shape) were markedly different between the normal epithelial and cancer cells in the mucosal surface. Kumagai et al. (2003) first reported an ex vivo study that confirmed the visualization of cells and nuclei in both normal esophageal squamous epithelium and esophageal squamous cell carcinoma (ESCC) [5] using a contact endoscope. Based on these initial investigations, Kumagai originally approached Olympus Medical Systems Co. to develop a flexible-type ultra-high-magnification endoscope that could be inserted into the digestive tract. A prototype EC was developed in 2003 as two types of probe that could be passed through the instrument channel of a thick mother endoscope (Table 1). Using this instrument, Kumagai et al. first reported their initial experience of visualizing human esophageal cancer lesions in vivo, clearly describing the characteristics of the surface epithelial cells of both normal squamous epithelium and esophageal cancer cells (Figure 1) [1]. On the basis of the knowledge acquired using the EC and further technical improvements, Olympus developed three other prototype ECs that overcame the respective problems of each preceding prototype. The latest prototype EC, the GIF-Y0074 (GIF-H290EC), was developed in 2015 [6]. This endoscope allows high-resolution observation under a high-definition view. Further, it offers continuous optical zooming up to ×500 using a hand lever, similar to magnifying endoscopes currently on the market. This EC is excellent in terms of ease of use, ready visualization of lesions, satisfactory resolution, and endoscope durability. Olympus launched this EC in 2018.

Among the various organs of the upper digestive tract, the esophagus has been studied most using the EC. Kumagai et al. [1,5] and Inoue et al. [7] were the first investigators in this field. Through a steady accumulation of cases, the EC was found to have high diagnostic accuracy for distinguishing benign from malignant lesions. The most notable feature of EC observation is that it can visualize surface epithelial cells, especially their nuclei. This facilitates final histological diagnosis and has the potential to obviate the need for biopsy histology. However, as the EC provides information on surface epithelial cells only, EC observation is not equivalent to conventional vertical histology. Therefore, EC diagnosis must be made by understanding the features of the conventional histology in each histological situation.

In this review, we describe the current status and future perspectives of EC observation of the esophagus.

## 2. Adequacy of Magnifying Power for Observation of the Esophagus

Histological diagnosis of esophageal cancer using the EC is based on observation of nuclear density and nuclear abnormality. For better recognition of nuclear shape, Kumagai et al. have recommended observation at a magnification of more than ×600. Using the third-generation EC (GIF-H290EC), the accuracy of diagnosis by a pathologist who was shown only photos of cells obtained using the EC was improved dramatically to 95% when digital magnification was introduced (increasing the power to more than x600) [8] relative to observation without digital magnification (81.5% at ×380) [9]. They reported that some cases of esophagitis showed a marked increase in nuclear density. In those cases, ×380 was insufficient for recognition of nuclear abnormality. Therefore, magnifying observation at more than ×600 is recommended to achieve a final histological diagnosis of esophageal lesions on the basis of nuclear shape. The latest EC model (GIF-H290EC) available commercially can magnify up to ×500 optically [6]. In this respect, the maximum magnifying power is still insufficient for recognition of nuclear abnormality. Digital zoom improves recognition of nuclear shapes and leads to better diagnostic accuracy.

## 3. Classification of Esophageal Squamous Epithelium

Two systems from different institutions have been proposed for the classification of squamous epithelium based on EC observation. Kawada et al. were the first to propose a three-tier classification of esophageal squamous epithelium visualized using the EC [10]: “non-malignant” (Type 1), “endocytoscopic borderline” (Type 2), and “malignant” (Type 3). Later, this classification was published as the “Type classification” by Kumagai et al. [11]. Inoue and colleagues later proposed that EC images of atypia could be classified into five tiers on the basis of nuclear density and nuclear abnormality (ECA classification) [12]. They also compared these findings with the Vienna classification, and found that the overall accuracy of EC for differentiating between non-malignant and malignant pathology was 82%. However, there has been some concern as to whether EC can distinguish all five different histological situations strictly, since it provides information on surface epithelial cells only. Recently, therefore, they proposed a three-tier modified EC classification (EC1: non-malignant, EC2: intraepithelial neoplasia (IN), and EC3: malignant), similar to the classification previously reported by Kumagai [13]. Although there are some differences in the interpretation of each category (especially for Type 2 and EC2), the EC classification of the esophageal epithelium has been unified into three tiers.

In 2022, based on their research experience of esophageal neoplasms and benign lesions, Kumagai et al. modified their type classification (Figure 2) [14], adding four non-malignant EC features in addition to the basic three-tier classification with reference to the pathologist’s opinion. They reported that their classification was helpful for making an accurate diagnosis of malignant and non-malignant lesions, even for non-expert endoscopists. In their classification, Type 1 corresponds to a non-malignant lesion that can be followed up without biopsy histology, and Type 2 is considered to be an EC-visualized borderline lesion. This category includes various pathological situations, such as regenerative squamous epithelium, squamous dysplasia, ESCC, etc., thus requiring biopsy histology for treatment planning. Type 3, characterized by a marked increase in nuclear density with prominent nuclear abnormality, is considered to be malignant histologically, without the need for confirmation by biopsy histology. Non-malignant EC features 1–4 correspond to the various types of esophagitis or presence of a white coat. However, if there is any possibility that such lesions could be malignant, confirmation using biopsy histology is recommended.

## 4. Principles of EC Observation and Appearance of Various Esophageal Lesions

As described above, EC provides information on cells and nuclear morphology at the mucosal surface because the dye used for vital staining can penetrate only one or two layers of the surface epithelium. This means that the findings of EC observation cannot be equivalent to those of conventional histopathology. Therefore, it is necessary to compare the surface morphology of cells visualized by EC with the findings of conventional vertical histopathology [15].

In histological sections, normal squamous cells show differentiation from the basal layer to the epithelial surface. The surface epithelial cells observed by EC are very thin and attenuated, resembling disks, and their nuclei are small and condensed. After vital staining, EC demonstrates homogeneously arranged superficial cells with nuclei showing regular characteristics of staining, shape, and dimension. The nuclear/cytoplasmic ratio is low (Type 1) (Figure 3a,b).

In histological sections of cancerous lesions, all layers of the squamous epithelium are rearranged by cancer cells in most cases, and the cancerous cells show massive aggregation with nuclei that are enlarged and irregular in size and shape. The nuclear/cytoplasmic ratio is high. These features can be clearly demonstrated by EC (Type 3) and are quite different from those of the normal epithelium (Figure 3c,d), being easily recognizable even for non-expert endoscopists.

Kumagai et al. reported the EC features of various types of esophagitis [14,16]. Gastro-esophageal reflux disease (GERD) shows a variety of EC findings. In the squamous epithelium beside the site of mucosal injury, circularly arranged squamous epithelial cells are observed around the epithelial papillae (“onion slice appearance”: non-malignant EC feature 1). Histologically, this EC finding reflects elongation of the epithelial papillae to the superficial layer due to GERD (Figure 3e,f). They also described that among cases of Grade C or D GERD, some were classified as Type 3 (neoplastic) by the endoscopist because of an obvious increase in nuclear density and prominent nuclear abnormality. Histologically, all of these cases were diagnosed as regenerative squamous epithelium. However, many regenerative epithelia show EC features that differ from those of carcinoma. First, clear cell borders can be recognized between the cells. Histologically, dilation of intercellular spaces (corresponding to the clear cell border demonstrated by EC) is reportedly one of the histological characteristics of GERD [17]. In addition, EC observation demonstrates some specific features of regenerative squamous epithelium: nuclear staining is weak, nucleoli are prominent, and the cytoplasm is stained more strongly than nuclei (non-malignant EC feature 2) (Figure 3g,h).

In GERD with erosions or post-irradiation esophagitis, the epithelium is absent. In some cases, spindle-shaped enlarged nuclei are demonstrated by EC. These are enlarged fibroblasts in the stroma. Histologically these features correspond to granulation or pseudo-malignant erosion (non-malignant EC feature 3) (Figure 3i,j).

EC observation of a white coat at sites of mucosal injury indicates aggregation of inflammatory cells, similar to that seen in esophagitis and esophageal cancer (non-malignant EC feature 4) (Figure 3k,l).

In cases of eosinophilic esophagitis, infiltration of inflammatory cells with bilobed nuclei (eosinophils) can be observed at the mucosal surface [16,18]. In cases of candida esophagitis, candida hyphae are sometimes evident [16].

## 5. EC Observation of Borderline Lesions (Squamous Intraepithelial Neoplasia (IN) and Squamous Dysplasia)

Pathological diagnosis of borderline lesions is known to be difficult and varies considerably among pathologists [19]. Furthermore, there are some differences in the definition of IN (squamous dysplasia) between the Japanese Esophageal Society classification [20] and the WHO classification [21]. High-grade dysplasia in the WHO classification includes the group of lesions also termed “carcinoma in situ” in Japan and other parts of Asia. Thus, there is a need to unify the pathological diagnosis of IN between Eastern and Western countries before the EC features of squamous IN can be discussed. Recently, Shimamura et al. tried to distinguish IN from non-cancerous epithelium and esophageal cancer [13]. They defined the EC features of IN as “slightly high nuclear density with a demarcation line, and centrally located round nuclei, which appear small or mildly enlarged”. However, based on the small number of IN cases they examined, the significance of their definition of IN is unclear [13,22]. Even if most IN cases were classified as EC2, biopsy diagnosis would be necessary to determine the treatment strategy, since EC2 includes various types of lesions, ranging from inflammation to cancer. Consequently, at the present time, the EC characteristics of IN remain to be determined. Further case series and histopathologically clear diagnostic criteria are needed.

## 6. Vital Dye Staining of Surface Epithelial Cells

For in vivo observation of cells using EC, vital staining is necessary. Among the dyes that are frequently used for endoscopic examination of the digestive tract, those that can stain the nucleus include methylene blue, toluidine blue, and crystal violet. Methylene blue and toluidine blue stain nuclei clearly, whereas crystal violet stains the cytoplasm rather than the nucleus. Kodashima et al. investigated the ideal staining times and concentrations of the various dyes [23]. They concluded that 1% methylene blue for 60 s provided optimal staining for the esophagus, and that 0.25% toluidine blue for 60 s was best for the stomach and colon. Minami et al. reported the usefulness of crystal violet and methylene blue double staining (CM double staining); a mixture of 0.1% methylene blue and 0.05% crystal violet was able to visualize surface epithelial cells more clearly than methylene blue staining alone [24].

However, Oliver et al. reported a potential risk of DNA damage when methylene blue staining was used under white light observation [25]. The National Toxicology Program (NTP) in the USA has reported that methylene blue shows some or equivocal evidence of carcinogenesis when force-fed to rats and mice [26]. Crystal violet is known to have carcinogenic properties, and the Canadian Endoscopic Society has prohibited its use in vivo. Although these dyes have been used widely in endoscopic examinations, no occurrence of cancer has ever been reported. For pathological diagnosis using EC, nuclear staining is essential, which means that vital staining is currently indispensable. However, the available evidence suggests that care should be taken to minimize the dose and select the right type of dye.

## 7. Merits of Toluidine Blue Single Staining

In consideration of the available evidence, Kumagai et al. have been using toluidine blue single staining for observation of the esophageal epithelium. Toluidine blue stains the nuclei of surface epithelial cells, and high-magnification EC demonstrates cancerous areas as dense aggregations of enlarged cancer cells, in contrast to the normal squamous epithelium. This means that, under non-magnified observation after toluidine blue staining, cancerous areas appear more darkly stained than the normal esophageal mucosa because of the high nuclear density, making the border between the two areas clearly visible. In some cases of esophageal cancer, no clear border can be visualized using iodine staining and narrow-band imaging (NBI). In such cases, EC observation using toluidine blue single staining can yield additional information for determining the border of ESCC (Figure 4).

## 8. AI Analysis for Images of Esophageal Cancer Obtained Using EC

Recently, artificial intelligence (AI) has made remarkable progress in various medical fields, especially as a system for screening medical images, including those in the field of radiological oncology [27], skin cancer classification [28], diabetic retinopathy [29], and histological classification of gastric biopsy samples [30]. Misawa et al. were the first to apply an AI system for analysis of endoscopic images. They reported that their system had high performance for distinguishing colon adenomas, and launched the world’s first commercial medical AI system [31]. Since then, in the field of colonoscopy, major endoscopy companies have also launched their own AI products [32]. Deep-learning analysis is the next step for texture-based analysis and has the potential to become a more powerful supportive tool for the interpretation of medical images based on historically accumulated sets of unique algorithms. Deep learning allows computational models, composed of multiple processing layers, to analyze representations of data with multiple levels of abstraction. Tada and colleagues have constructed a deep learning AI-based diagnostic system involving a convolutional neural network (CNN). They successfully applied this AI system for the diagnosis of *Helicobacter pylori* infections in the stomach [33] and the detection of gastric cancer [34].

In 2019, Horie et al. reported a first-generation AI for the early detection of esophageal cancer [35]. Although it showed high sensitivity for the detection of esophageal cancer (98%), its positive predictive value was low (40%). Their AI misdiagnosed shadows, and normal esophageal structures, as esophageal cancer because their test image set did not include any normal esophageal pictures. Thereafter, several AIs were trained with larger numbers of endoscopic pictures obtained from both normal esophagus and esophageal cancer. This has led to a marked improvement in diagnostic performance, exceeding the results of expert endoscopists in terms of not only early diagnosis, but also characterization (diagnosis of tumor invasion depth and benign/malignant differentiation) [36,37,38,39,40,41].

Kumagai et al. created an AI for the analysis of EC images of esophageal lesions. Their first AI for evaluation of such images using third and fourth-generation EC was reported in 2019 [42]. They trained their AI with 4715 pictures of both malignant and non-malignant lesions, and tested 1520 EC pictures. This facilitated extremely high-speed analysis (0.02 s per image), and showed high diagnostic performance (accuracy: 90.9%). In 2022, they reported the diagnostic performance of their second-generation AI trained using a larger number of images (7983) using the fourth-generation EC currently on the market [14]. They tested 114 randomly arranged EC images obtained from both malignant and non-malignant lesions. Surprisingly, the diagnostic accuracy of the AI was equal to that of the expert pathologist (91.2%), who had reviewed more than 700 EC images obtained using the first to fourth-generation EC models.

As described above, research on EC began in 2000 by ourselves and became commercially available in 2018. EC is capable of observing cellular morphology and arrangements observed from the mucosal surface in vivo in real time that reflects conventional histopathology. Based on the EC images with maximum magnification, we believe that the need for biopsy histology will be reduced. In the near future, it is expected that AI will support endoscopists for the early detection of cancer using non-magnified observation, diagnosis of invasion depth using magnified observation, and final histological diagnosis using EC in vivo.

## Figures and Tables

**Figure 1 diagnostics-12-02222-f001:**
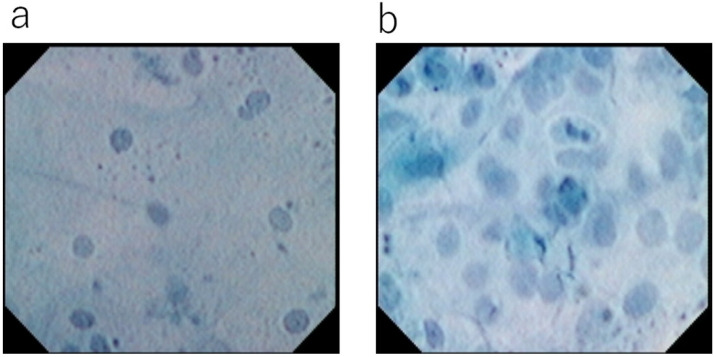
Ultra-high magnifying observation using 1st generation EC with methylene blue staining (XEC120U 1125x Olympus Medical systems Co., Tokyo, Japan): (**a**) Normal esophageal mucosa. Surface epithelial cells have a low nuclear/cytoplasm ratio and no nuclear abnormality. (**b**) Esophageal squamous cell carcinoma. Cancerous cells show an increase in nuclear density in comparison with the normal squamous epithelium and have prominent nuclear abnormalities.

**Figure 2 diagnostics-12-02222-f002:**
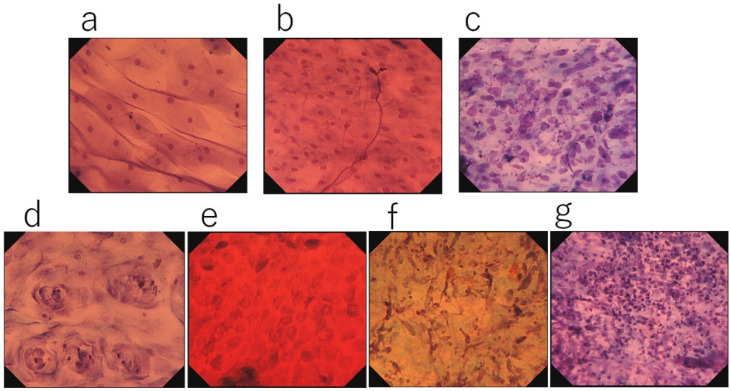
Type classification: (**a**) Type 1, surface epithelial cells show a low nucleus/cytoplasm (N/C) ratio and no nuclear abnormality. Nuclear density is low or slightly increased (toluidine blue staining, ×900). (**b**) Type 2, evident increase in nuclear density. Slight nuclear abnormality is evident, but not sufficiently obvious to be considered type 3 (toluidine blue staining, ×900). (**c**) Type 3, evidently increased nuclear density and nuclear abnormality, e.g., irregular nuclear size and shape, with hyperchromatism (toluidine blue staining, ×900). (**d**) Non-malignant EC feature 1 (GERD): circularly arranged squamous cells without nuclear abnormality around epithelial papillae (toluidine blue staining, ×900). (**e**) Non-malignant EC feature 2 (regenerative squamous epithelium): evident increase in nuclear density with differing nuclear sizes and shapes. Densely stained cytoplasm with a clear cell border. Nuclei are stained weakly, and multiple nucleoli are evident (toluidine blue staining, ×900). (**f**) Non-malignant EC feature 3 (erosion, granulation): increase in nuclear density with spindle-shaped enlarged nuclei (enlarged fibroblasts in the stroma, toluidine blue staining, ×900). (**g**) Non-malignant EC feature 4 (white coat): evident aggregation of inflammatory cells (toluidine blue staining, ×500).

**Figure 3 diagnostics-12-02222-f003:**
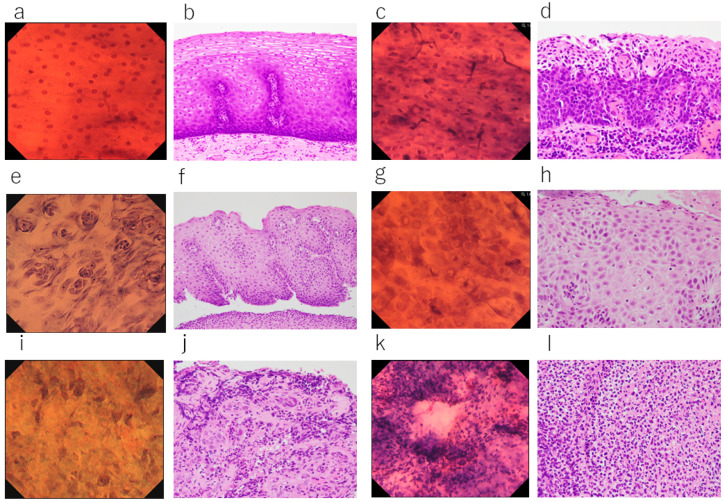
Comparison between EC observations and histological findings: (**a**) EC image of normal esophageal mucosa (Type 1, toluidine blue stain, ×900). (**b**) Histological section of normal esophageal mucosa (HE, ×400). (**c**) EC image of esophageal squamous cell carcinoma. Increased nuclear density, nuclear abnormality, and nuclear enlargement are evident (Type 3, toluidine blue stain, ×900). (**d**) Histological section of esophageal squamous cell carcinoma (histological section of the same lesion as (**c**)). Atypical cells have replaced the entire epithelial layer (HE, ×400). (**e**) EC image of esophagitis. Non-malignant cells are circularly arranged around the epithelial papilla (non-malignant EC feature 1, toluidine blue stain, ×500). (**f**) Histological section of the same area as (**e**). Elongation of the epithelial papillae is evident just beyond the epithelial surface (HE, ×200). (**g**) EC image of regenerating squamous epithelium. Weakly staining nuclei and prominent nucleoli are evident. The cytoplasm is darkly stained and intercellular space dilation is clearly recognized (non-malignant EC feature 2, toluidine blue stain, ×900). (**h**) Histology of the same area as (**g**). Prominent nucleoli are evident within the nuclei, with clear intercellular borders (HE, ×400). (**i**) EC image of esophageal erosion. Increased nuclear density, nuclei of variable size, and spindle-shaped enlarged nuclei are evident (non-malignant EC feature 3, toluidine blue stain, ×900). (**j**) Histological section of the same area as (**i**). Inflammatory cell infiltration and stromal fibroblasts are observed as nuclear atypia (HE, ×400). (**k**) EC image showing the white coat of radiation esophagitis. Dense aggregation of small inflammatory cells is evident (non-malignant EC feature 4, toluidine blue stain, ×900). (**l**) Histopathological image of the same area as (**k**), showing dense aggregation of inflammatory cells (HE, ×400).

**Figure 4 diagnostics-12-02222-f004:**
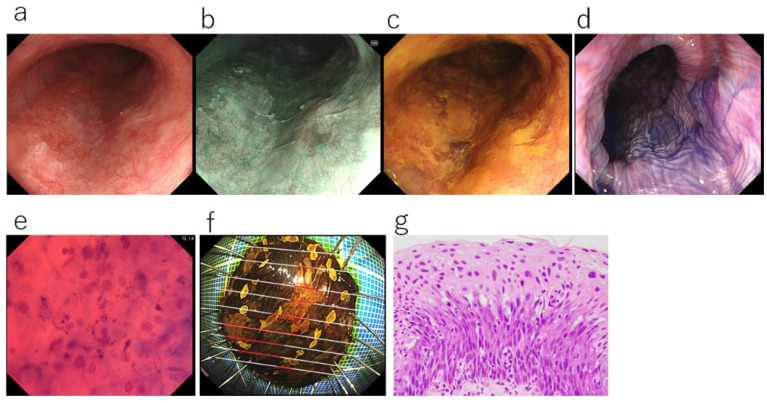
A case in which toluidine blue staining was useful for diagnosing the extent of esophageal cancer: (**a**) White light observation. Whitish and rough mucosa is evident, but the boundary of the esophageal carcinoma is unclear. (**b**) Narrow-band imaging (NBI). The lesion is not delineated as a brownish area and its boundary is unclear. (**c**) Iodine staining. The lesion is a mixture of unstained and weakly stained areas, and the boundary is also unclear. (**d**) Toluidine blue staining shows the lesion as a dark stained area with a well-demarcated border. (**e**) Ultra-high-magnification observation using the EC (toluidine blue staining, ×900). The area darkly stained with toluidine blue was confirmed to be esophageal carcinoma because of increased nuclear density with nuclear abnormality (nuclear atypia). (**f**) Resected specimen after endoscopic treatment (iodine staining). Unstained and weakly stained areas are intermixed. The cancerous area (red line) matches the area darkly stained with toluidine blue. (**g**) The histological diagnosis was squamous cell carcinoma in situ (0-IIb, T1a-EP, HE staining, ×400).

**Table 1 diagnostics-12-02222-t001:** Four generations of endocytoscope.

	XEC120U 1st Generation	XEC300F 1st Generation	GIF-Y0001 2nd Generation	GIF-Y0002 3rd Generation	GIF-H290EC 4th Generation
**Type of scope**	probe	probe	scope (2 lens integrated)	scope	scope
**Magnification power**	×1125	×450	×450	Optical: ×380	Optical: ×500
				With electrical: ×700	With electrical: ×900
**Observation field**	120 μm × 120 μm	300 μm × 300 μm	400 μm × 400 μm	Optical: 700 μm× 600 μm	Optical: 570 μm × 500 μm
				With electrical: 440 μm × 380 μm	With electrical: 360 μm × 310 μm
**Magnification method**	fix	fix	fix	Continuous zoom	Continuous zoom
**Outer diameter**	3.4 mm	3.4 mm	11.6 mm	10.7 mm	9.7 mm

## Data Availability

Data sharing not applicable.

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
