# Peer review of "Endocytoscopic Observation of Esophageal Lesions: Our Own Experience and a Review of the Literature"

_diagnostics, 2022, doi:10.3390/diagnostics12092222_

Round 1

Reviewer 1 Report

The authors present a brief but comprehensive review on the use of an endocytoscope with vital staining in the risk stratification of esophageal lesions. They review the classification schemes most useful to differentiate between malignant and benign. I have several questions in regards to the utilization of the endocytoscope in daily practice.

1. Do the authors propose that biopsy is unnecessary for the lesions designated benign (EC1) and if so, what follow-up exists for that recommendation?

2. Do the authors foresee a time when biopsy/histology will be unnecessary for EC2 lesions?

3. Do the authors currently diagnose malignancy using the endocytoscope only? Or are biopsies still being taken in daily practice? If biopsies are still being taken, at what point will the authors be comfortable not taking tissue for histology? Or if biopsies are no longer being taken, how many EC3 samples were taken before that decision was made?

Author Response

I attach the cover letter.

Reviewer 2 Report

Overall, the review is well-written, clear, and informative. It describes capabilities and addresses limitations of the technique.

Minor:

To make it a bit easier to follow, for figure 3 legend, identify by letter which subfigures are images of the same area as a previous one. For example, F: histological section of the same area as E. 

Author Response

I attach the cover letter.
